# Lactoferrin-Anchored Tannylated Mesoporous Silica Nanomaterials-Induced Bone Fusion in a Rat Model of Lumbar Spinal Fusion

**DOI:** 10.3390/ijms242115782

**Published:** 2023-10-30

**Authors:** Sung Hyun Noh, Kanghyon Sung, Hye Eun Byeon, Sung Eun Kim, Keung Nyun Kim

**Affiliations:** 1Department of Neurosurgery, Yonsei University College of Medicine, 50, Yonsei-ro, Seodaemun-gu, Seoul 03722, Republic of Korea; juwuman12@gmail.com; 2Department of Neurosurgery, Ajou University School of Medicine, 206, World cup-ro, Yeongtong-gu, Suwon-si 16499, Republic of Korea; 3Department of Orthopedic Surgery, College of Medicine, Korea University, 73, Korea-daero, Seongbuk-gu, Seoul 02841, Republic of Korea; ns.khsung@gmail.com; 4Institute of Medical Science, Ajou University School of Medicine, 206, World cup-ro, Yeongtong-gu, Suwon-si 16499, Republic of Korea; 110236@aumc.ac.kr; 5Department of Orthopedic Surgery and Nano-Based Disease Control Institute, Korea University Guro Hospital, 148, Gurodong-ro, Guro-gu, Seoul 08308, Republic of Korea; 6Department of Neurosurgery, Spine and Spinal Cord Institute, Severance Hospital, Yonsei University College of Medicine, 50, Yonsei-ro, Seodaemun-gu, Seoul 03722, Republic of Korea

**Keywords:** lactoferrin, bone fusion, rat, spine, nanoparticles

## Abstract

Lactoferrin (LF) is a potent antiviral, anti-inflammatory, and antibacterial agent found in cow and human colostrum which acts as an osteogenic growth factor. This study aimed to investigate whether LF-anchored tannylated mesoporous silica nanomaterials (TA-MSN-LF) function as a bone fusion material in a rat model. In this study, we created TA-MSN-LF and measured the effects of low (1 μg) and high (100 μg) TA-MSN-LF concentrations in a spinal fusion animal model. Rats were assigned to four groups in this study: defect, MSN, TA-MSN-LF-low (1 μg/mL), and TA-MSN-LF-high (100 μg/mL). Eight weeks after surgery, a greater amount of radiological fusion was identified in the TA-MSN-LF groups than in the other groups. Hematoxylin and eosin staining showed that new bone fusion was induced in the TA-MSN-LF groups. Additionally, osteocalcin, a marker of bone formation, was detected by immunohistochemistry, and its intensity was induced in the TA-MSN-LF groups. The formation of new vessels was induced in the TA-MSN-LF-high group. We also confirmed an increase in the serum osteocalcin level and the mRNA expression of osteocalcin and osteopontin in the TA-MSN-LF groups. TA-MSN-LF showed effective bone fusion and angiogenesis in rats. We suggest that TA-MSN-LF is a potent material for spinal bone fusion.

## 1. Introduction

Spinal fusion is an increasingly common procedure used to treat various pathologies arising from trauma, degenerative diseases, infections, and tumors [1]. However, pseudarthrosis can occur because of a failure of bone fusion, resulting in pain, instability, and disability [2]. Failure of bone bridging increases the risk of device failure, with sedimentation or screw loosening [3]. Pseudarthrosis rates range from 5% to 34% [4]. Successful bone fusion after spinal fusion is challenging, particularly in elderly patients with osteoporosis, due to low bone strength and poor bone quality. Over the past decade, a wide range of treatment modalities to prevent fusion failure or pseudarthrosis have been explored, with research focusing on the identification of new substances that aid fusion.

Employing an autograft iliac crest bone is the clinical “gold standard” in fusion procedures. However, these autografts have several disadvantages. Collecting a consistent volume of the iliac crest bone may be difficult, and the harvesting procedure can be painful and may be associated with hematomas, infections, fractures at the donor location, and prolonged surgical time. Furthermore, there are limitations in the use of bone autografts with regards to patients with osteoporosis or metabolic diseases.

Except for autologous bone graft material, spine surgeons have used various methods, including allogeneic bone graft materials (e.g., demineralized bone allograft), synthetic bone graft materials (e.g., tricalcium phosphate, hydroxyapatite, bioglass), and bone morphogenetic protein-2 (BMP2) to achieve bone fusion [5]. BMP2 was approved in 2002 and is currently used for spinal fusion. Although BMP2 is effective in osteogenic differentiation and osteogenesis, delivery of BMP2 alone is not effective in supporting osteogenesis due to its short half-life. BMP2 diffuses rapidly throughout body fluids and is cleared rapidly [6]. Therefore, higher than physiological doses of BMP2 have been used clinically to promote bone formation. However, high doses of BMP2 are associated with serious complications such as soft tissue inflammation and heterotopic ossification [7].

Lactoferrin (LF) is an 80 kDa non-heme iron-binding protein. produced primarily by exocrine epithelial cells and expressed in most tissues, that plays an important role in the innate immune system of mammals [8]. LF is a potent antiviral, anti-inflammatory, and antibacterial agent found in cow and human colostrum [9]. Moreover, LF acts as an osteogenic growth factor. A previous report showed that treatment with LF not only increased osteoblast proliferation but also promoted osteoblast differentiation [8]. Previous studies examined the oral administration or local injection of LF [10,11,12]. Yanagisawa et al. found that oral administration of LF suppressed the progression of rheumatoid arthritis in a mouse model [10]. Guo et al. observed that oral administration of LF to ovariectomized rats resulted in preserved bone mass but also improved bone microarchitecture [11]. Furthermore, Cornish et al. reported that local injection of LF over the hemicalvaria increases bone growth in adult male mice [12].

A previous study examined LF-anchored tannylated mesoporous silica nanomaterials (TA-MSN-LF) [13]. LF is a promising protein for osteogenic differentiation and tendon healing; however, without the assistance of a delivery medium, the effect of the protein drug was not as effective as expected because of its short half-life in the blood [14]. Therefore, recent research has focused on the use of MSN and TA. MSN are attracting considerable attention for their potential in biomedical applications because they have excellent biocompatibility and biodegradability compared with other inorganic nanomaterials [15]. Further advantages of MSN from a bone tissue engineering perspective are their large surface area, large pore size, and easy surface modification [16]. TA functions as both an organic and inorganic sample due to the presence of hydroxyl and galloyl groups [17]. TA is known to interact directly with several types of biopolymers (DNA, gelatin, collagen, albumin, chitosan, thrombin, enzymes) because of its electrostatic interaction, hydrogen bonding, and hydrophobic interaction abilities [18]. The osteo-differentiation ability of TA-MSN-LF has been demonstrated in vitro [13]. This study aimed to investigate whether TA-MSN-LF functions as a bone fusion material in a rat model of lumbar spinal fusion. This is the first study to demonstrate the effectiveness of LF in an animal model of lumbar spinal fusion.

## 2. Results

Of the 33 rats, 3 died during surgery due to side effects of general anesthesia, and 30 survived without any complications. The behavior of the animals showed no specific changes suggesting side effects of the chemicals or local complications such as nerve irritation. Finally, the lumbar fusion models used in the analysis were as follows: TA-MSN-LF high group, 8; TA-MSN-LF low group, 8; MSN group, 8; and defect group, 6.

### 2.1. Manual Palpation

Table 1 shows the qualitative fusion assessment by manual palpation at 8 weeks after fusion. Qualitative parameters such as degree of fusion, degree of hardness, and volume of the bridged bone were significantly different between each group (*p* < 0.001). Additionally, the manual qualitative fusion scores were significantly different between each group (*p* < 0.001). All parameters derived from manual palpation tended to show better results in the TA-MSN-LF high group than in the other groups. These results demonstrated that promotion of bone fusion was superior in the TA-MSN-LF high group.

### 2.2. Micro-CT

Table 2 shows the quantitative results of the micro-computed tomography (CT) findings 8 weeks after fusion. Micro-CT of the TA-MSN-LF high and low groups revealed new bone formation in the decortication area (Figure 1). Additionally, quantitative fusion bone volume and trabecular bone number were significantly different among the groups (*p* < 0.05). Bone volume and trabecular number in the TA-MSN-LF high group were higher than those in the other groups. However, there were no significant differences in trabecular bone thickness and trabecular separation among the four groups. These results demonstrated that promotion of bone fusion was superior in the TA-MSN-LF high group.

### 2.3. Histologic Results

Hematoxylin eosin (H&E) and osteocalcin (OCN) staining were performed to evaluate new bone formation 8 weeks after the procedure. As shown in Figure 2, histological sections were stained with H&E and the boxes indicate new bone formation in the TA-MSN-LF high and low groups. Figure 3A shows staining for OCN, a bone formation marker, in each group. Staining was best performed in the TA-MSN-LF-high group. Figure 3B quantifies this and compares the osteocalcin levels. The TA-MSN-LF high and low groups had significantly higher OCN intensity than that in the other groups (*p* < 0.05). The OCN intensity of the TA-MSN-LF-high group was significantly higher than that in the TA-MSN-LF-low group (*p* < 0.05). Moreover, the blood vessel frequency in the TA-MSN-LF high group was significantly higher than that in the other groups (*p* < 0.05, Table 3). The histologic results indicate that bone fusion and angiogenesis were most effective in the TA-MSN-LF high group.

### 2.4. Bone Turnover Marker

The serum OCN concentration was significantly higher in the TA-MSN-LF high and low groups than that in the other groups (*p* < 0.05, Figure 4). The serum OCN concentration of the TA-MSN-LF-high group was significantly higher than that in the TA-MSN-LF-low group (*p* < 0.05). The serum bone turnover marker results show that bone fusion ability was the highest in the TA-MSN-LF high group.

### 2.5. Quantification of Osteo-Differentiation-Specific Genes

The mRNA expression levels of OCN and osteopontin (OPN) were significantly higher in the TA-MSN-LF high and low groups than those in the other groups (*p* < 0.05, Figure 5). The mRNA expression level of OCN and OPN of the TA-MSN-LF-high group was significantly higher than that in the TA-MSN-LF-low group (*p* < 0.05). This finding may indicate that TA-MSN-LF-high may promote and accelerate bone formation.

## 3. Discussion

Although the development of new substances for bone tissue regeneration remains challenging, numerous biomaterials have been proposed for this purpose. LF is an osteoinductive factor that can promote the osteogenic differentiation of various cells, such as osteoblasts, C2C12 cells, and adipose-derived stem cells (ADSC) [19]. In a previous study, we developed a new type of TA-MSN-LF to release LF from nanoparticles over a long period of time and impart osteoconductive effects to the silica nanoparticles [13]. In this study, this material was placed into rats to determine its effectiveness as a bone fusion material.

LF is a widely distributed glycoprotein in mammals; human tears contain approximately 1.13 g/L of LF and human colostrum contains more than 5 g/L [20,21]. LF is a crucial anabolic bone growth factor and can function as an effector molecule for bone remodeling [22]. In vitro experiments have shown that local distribution of LF strongly increased the proliferation and differentiation of osteoblasts from mice to humans [23,24]. Other studies have demonstrated that the oral administration of LF not only improves bone formation and sclerosis in distraction osteogenic animal models, but also improves bone mineral density (BMD) and biomechanical strength in ovariectomized animal models [25,26].

To construct the TA-MSN-LF, LF was immobilized on the MSN surface coated with TA. TA is a representative molecule with strong affinity for various proteins, such as proline-rich ones [27]. Each MSN group was observed by transmission electron microscope (TEM), and dynamic light scattering (DLS) was performed in the range of 260–300 nm. We also used X-ray photoelectron spectroscopy (XPS) to investigate the surface chemical composition of MSNs containing TA and/or LF and compared this with the composition of pure MSNs. TA-MSN-LF exhibited a sustained release of LF for up to 28 days; these results demonstrated that TA-MSN-LF nanoparticles can achieve long-term delivery of LF. Park et al. investigated whether local bone regeneration occurred in a rat calvarial defect model using LF and poloxamer gels [28], which were used as the LF carriers. However, LF was released for up to three days under these conditions. In conclusion, the LF/poloxamer material increased cell viability and was less likely to induce immune responses; however, the formulation failed to enhance bone regeneration in vivo. Conversely, the role of the TA-MSN as a transmitter was significant in our study.

In this study, we evaluated whether TA-MSN-LF could induce the formation of bone tissue and spinal regeneration in vivo as an alternative to autologous bone.

In a rat model, TA-MSN-LF significantly increased new bone tissue formation and vertebral bone regeneration 8 weeks after a spinal fusion procedure; additionally, the bone volume and trabecular number were significantly higher in the high-dose TA-MSN-LF group (*p* < 0.05). Guo et al. analyzed the effects of oral LF on bone mass and microarchitecture in a rat model of osteoporosis [11]. The LF dose was divided into three groups: 0.85 mg/kg body weight, 8.5 mg/kg body weight, and 85 mg/kg body weight. Micro-CT analysis revealed that bone volume was the highest in the 85 mg/kg body weight group. Koca et al. investigated whether LF was beneficial for autograft healing in peri-implant bone in a rat model [29]; the percentage of bone volume (bone volume/total volume) in the micro-CT analysis was significantly higher in the defect filled with autograft and LF (100 µg/mL) group compared with that in the autograft-only group. Thus, TA-MSN-LF conclusively induces bone tissue formation and spinal regeneration.

Park et al. analyzed the effects of LF in a rat model of calvarial defects [28]. Histologic analysis evaluating the osteogenic effects of LF revealed that bone defects were reduced in the group injected with LF compared with those in the control group (*p* < 0.05). Guo et al. reported that a group treated with LF presented higher serum OCN levels than those in the control group [11]. The authors examined osteoblast-like cell metabolic activity at 24 h, which increased in the group that consumed LF. In the present study, OCN was stained and quantified. In the TA-MSN-LF high and low groups, OCN staining was good, and the OCN intensity was significantly higher than that in the other groups. Additionally, the mRNA expression levels of OCN and OPN were significantly higher in the high and low TA-MSN-LF groups than those in the other groups (*p* < 0.05). As a biologically active molecule, LF promotes proliferation and differentiation of various cells [19]. It also inhibits osteoclast formation by reducing the number of osteoclasts that can actively resorb bone [30]. These positive effects of LF support the use of LF in bone tissue regeneration as very interesting. However, due to the low bioavailability of LF in vivo, nanomaterial-based strategies have been developed to improve the biological activity of LF. This in vitro study reported that TA-MSN-LF promotes osteogenesis and angiogenesis. Although this study did not compare TA-MSN-LF with other materials, Chang et al. reported that 100 µg of LF produced more substantial osteoinductive effects than BMP-2 [31], concluding that LF could replace BMP-2 [31]. Li et al. studied the osteoblast differentiation ability of lactoferrin and reported that 100 µg was most effective [32]. And Zhang et al.’s study also reported that 100 µg of lactoferrin was most effective in proliferative activity of the osteoblast cells [33]. Considering the importance of bone fusion ability, the TA-MSN-LF high (100 µg/mL) was excellent in inducing differentiation and promoting proliferation in bone tissue engineering in our study.

This study had several limitations. First, the sample size was small (eight rats in each group). However, this is the first experimental report confirming the effect of LF in a spinal fusion model. Second, because the effect of LF was evaluated at a single time point, the degree of healing could not be assessed in all groups over various time periods. Third, because only one graft material was used, we were unable to evaluate the effect of LF on bone healing using other graft materials (e.g., allografts). Nevertheless, to the best of our knowledge, this is the first study to demonstrate that LF induces effective bone fusion in a spinal fusion model. Further studies are required to elucidate the systemic side effects and local complications and to establish safe and effective doses of LF. Clinical trials are required to demonstrate the clinical outcomes of LF.

## 4. Materials and Methods

### 4.1. Animals

All animal procedures were conducted according to the guidelines of the Laboratory Animal Center (Korea-2020-0059-C1). The experimental animals were housed in a specific-pathogen-free facility and fed a standard diet. All animals used in this study were female Sprague Dawley (SD) rats (weight, 100–180 g) aged 6–9 weeks.

### 4.2. Materials

Mesoporous silica nanomaterials (MSNs) (Sigma-Aldrich, St. Louis, MO, USA) were modified with tannic acid (TA) (Sigma-Aldrich) to immobilize human LF (Sigma–Aldrich, St. Louis, MO, USA). First, 10 mg of MSN was added to a phosphate-buffered saline (PBS) solution (pH, 7.4) containing dissolved TA (concentration, 50 µg/mL) and the mixture was gently shaken overnight at room temperature (RT). MSNs containing TA were then washed twice with distilled water (DW) and lyophilized for two days. The TA load on MSN was confirmed by analyzing the amount of residual TA in the PBS solution. Hereafter, the MSNs with TA are referred to as TA-MSNs.

To immobilize LF (concentration, 1 or 100 µg/mL) on the MSN surface, TA-MSN (concentration, 10 µg/mL) and LF (concentration, 1 or 100 µg/mL) were added to the PBS solution and then incubated for 24 h. Next, all samples were washed three times with DW at 3000 rpm and 4 °C for 10 min using a Smart R17 Centrifuge (Hanil Science Industrial, Incheon, Republic of Korea). The samples were freeze dried for two days.

To assess the LF load, the supernatant was collected after the immobilization of LF on TA-MSN and analyzed using the Pierce Bicinchoninic Acid (BCA) Protein Assay Kit (Thermo Fisher Scientific, Rockford, IL, USA), according to the manufacturer’s protocol. The LF (1 µg/mL) anchor-type TA-MSN and LF (100 µg/mL) anchor-type TA-MSN are hereafter referred to as TA-MSN-LF low and TA-MSN-LF high, respectively. A total of 3 mg was used in the experiment, and the real LF dose was calculated when 3 mg of TA-MSN-LF was added, which is 0.03 µg/rat (1 µg/mL TA-MSN-LF) and 2.75 µg/rat (100 µg/mL TA-MSN-LF). The use of TA-MSN-LF low (1 µg/mL) and TA-MSN-LF high (100 µg/mL) in our study was based on previous studies using LF [31,32,33]. Chang et al. evaluated the osteo-differentiation of adipose-derived stem cells LF dose (10 µg/mL, 20 µg/mL, 50 µg/mL, 100 µg/mL, and 500 µg/mL) and concluded that 100 µg/mL was the most effective dose [31]. In Li et al.’s study, lactoferrin doses of 1 µg, 10 µg, 100 µg, and 1000 µg were studied. And in a study by Zhang et al., doses of lactoferrin of 1 µg, 10 µg, and 100 µg were studied. Therefore, in the present study, we compared 1 µg/mL and 100 µg/mL doses.

### 4.3. Experimental Design and Surgical Procedure

A single-level bilateral lumbar posterolateral fusion was performed, and thirty host rats were divided into four experimental groups: [A] LF (100 µg/mL) anchor-type TA-MSN (*n* = 8), [B] LF (1 µg/mL) anchor-type TA-MSN (*n* = 8), [C] MSN (*n* = 8), and [D] defect (*n* = 6). After anesthesia with a mixed solution of zoletil–xylazine (zoletil at 20 mg/kg and xylazine at 10 mg/kg), the hair on the surgical site was shaved. The vertebral levels from L4 to L5 were identified by palpation and anatomical landmarks. A dorsal midline skin incision was made at the center of the L4–L5 spinous processes, and the edges of the skin were retracted using a self-holding aspirator. An intermuscular plane was established between the multifidus and longissimus muscles, thereby exposing the transverse process from L4 to L5. Decortication of the transverse processes and external segments/intervertebral joints was performed using an electric bar. After decorticating the transverse processes on both sides of L4 and L5, 3 mg of each material was implanted on each side of the fusion bed space (L4–L5). After suturing the muscle and skin layer by layer, cefazolin (100 mg/kg) was injected. The rats were euthanized eight weeks after the experiment with an anesthetic gas. Figure 6 illustrates the experimental animal process.

### 4.4. Manual Palpation

Eight weeks after the fusion surgery, the lumbar spine specimens were carefully collected, and the soft tissue around the specimens was carefully removed. To evaluate the degree of fusion, the qualitative degree of fusion, degree of hardness, and amount of bridged bone were investigated. Flexion, extension, side flexion, and torsion forces were applied and the qualitative fusion grade was measured as follows: Grade (1), a nonunion state in which the range of motion of the index part presented as that of a normal lumbar spine; Grade (2), a fused state in which the range of motion of the index part was reduced but presented some flexibility; Grade (3), a state in which the range of motion of the index part was absent, the motion in the stationary state was smooth, and the hardness was similar to that of a hard bone. The degree of hardness of the fused portion was examined using direct finger pressure and classified according to the following subjective judgment: Grade (1), no or minimal hardness; Grade (2), hard but not satisfying hard bone; and Grade (3), hard bone-like absolute hardness. The volume of the bridging bone between the transverse processes was evaluated using direct visual inspection and finger palpation and was classified as follows: Grade (1), no or minimal volume of bridging bone; Grade (2), flat bone-like apparent volume; and Grade (3), flower-like excess volume compared with flat bone. Passive qualitative fusion scores were rated from 0 to 9 according to the sum of all grades obtained by manual palpation.

### 4.5. Micro CT

All samples were scanned using a high-resolution micro-CT (SkyScan-1072, Skysan, Kontich, Belgium) running at 90 kV and 200 mA. SkyScan1173 control software (Ver 1.6, Bruker-CT, Billerica, MA, USA) was used for the measurement, with a tube voltage of 130 kVp, a tube current of 60 µA, 1 mm aluminum filtration (Filter), exposure time of 500 ms, (2240 × 2240) pixels, and pixel size of 35.00 µm. Rotation angles of 0.3° and 180° were used to obtain 800 high-resolution images. For cross-sectional reconstruction, an image of 2240 × 2240 pixels was obtained using Nrecon (Ver 1.7.0.4, Bruker-CT, Billerica, MA, USA), and the cross-sectional image was aligned using Dataviewer (Ver.1.5.1.2, Bruker-CT, Billerica, MA, USA). For data analysis, the area was set using CTAn (Ver 1.17.7.2, Bruker-CT, Billerica, MA, USA), and the volume of the new bone area was analyzed by setting the threshold to 90–255 for the amount of bones present and bone parameters in the area. For quantitative fusion analysis, the bone union volume between the transverse processes was analyzed. Scans of the newly formed bone were evaluated from the lower endplate of the lower spine to the upper endplate of the upper spine.

### 4.6. Histologic Evaluation

After all radiographic examinations, all spinal specimens were desalted in 10% formaldehyde solution for seven days and embedded in paraffin wax. Serial sections (thickness, 5 mm) through the largest defect diameter were stained with H&E to identify cellular responses indicative of osteogenesis. Serial sections (thickness, 5 mm) up to the maximum defect diameter were stained with H&E to evaluate cellular responses indicative of bone formation. Immunohistochemical analysis of OCN expression was performed. We measured OCN intensity and analyzed the quantity of blood vessels using a semi-quantitative histological evaluation. Blood vessel frequency was determined using the following scoring criteria: 1 = often; 2 = quite often; 3 = frequent; and 4 = very frequent.

### 4.7. Bone Turnover Marker

The quantitative osteogenic ability of each group to induce osteoblast activity was compared by evaluating serum OCN levels using an OCN ELISA kit (Novus, San Diego, CA, USA). Eight weeks after surgery, blood was collected from the tail vein under zoletyl-xylazine anesthesia. All standards and samples were analyzed using an iMark microplate reader (Bio-Rad, Hercules, CA, USA) at a wavelength of 450 nm.

### 4.8. Quantification of Osteo-Differentiation-Specific Genes

To further confirm the osteogenic capacity of MSNs with or without LF, the mRNA levels of osteogenic differentiation-specific genetic markers such as OCN and OPN were examined using real-time polymerase chain reaction (PCR). Eight weeks after surgery, blood was collected from the tail vein under zoletyl–xylazine anesthesia. Total RNA was purified using A TRIzol reagent and cDNA was prepared from total RNA (1 µg) using a PrimeScriptTM 1st Strand cDNA Synthesis Kit (Takara Bio Inc., Ostu, Japan). PCR amplification and real-time PCR were performed using an ABI7300 Real-Time Thermal Cycler (Applied Biosystems, Foster City, CA, USA). OCN and OPN genes were normalized to glyceraldehyde 3-phophate dehydrogenase (GAPDH). All experiments were repeated three times at each time point. Expression levels were calculated using the 2^−ΔΔCT^ method.

### 4.9. Statistical Analysis

All statistical analyses were performed using SPSS 25 version (SPSS Inc., Chicago, IL, USA). Results were presented as mean values ± standard deviation. Depending on the nature of the parameters, Pearson’s chi-square test, the nonparametric Wilcoxon rank-sum test, and centralized analysis of variance (ANOVA) were used. Statistical significance was set to a *p* value < 0.05.

## 5. Conclusions

TA-MSN-LF possessed effective bone fusion and angiogenesis effects in rats, which suggests that TA-MSN-LF is a potent material for spinal bone fusion. In particular, the bone fusion ability of TA-MSN-LF high (100 µg/mL) was higher than that of TA-MSN-LF low (1 µg/mL).

## Figures and Tables

**Figure 1 ijms-24-15782-f001:**
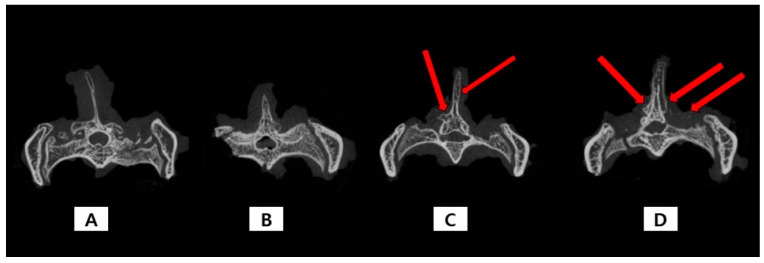
The representative cases of micro-CT: (**A**) Micro-CT of the defect group shows no new bone formation in decortication area; (**B**) Micro CT of the MSN group shows no new bone formation in the decortication area; (**C**) Micro-CT of the TA-MSN-LF low group shows some bone formation in the decortication area (red arrow); (**D**) Micro-CT of the TA-MSN-LF high group shows a large area of bone formation in the decortication area (red arrow).

**Figure 2 ijms-24-15782-f002:**
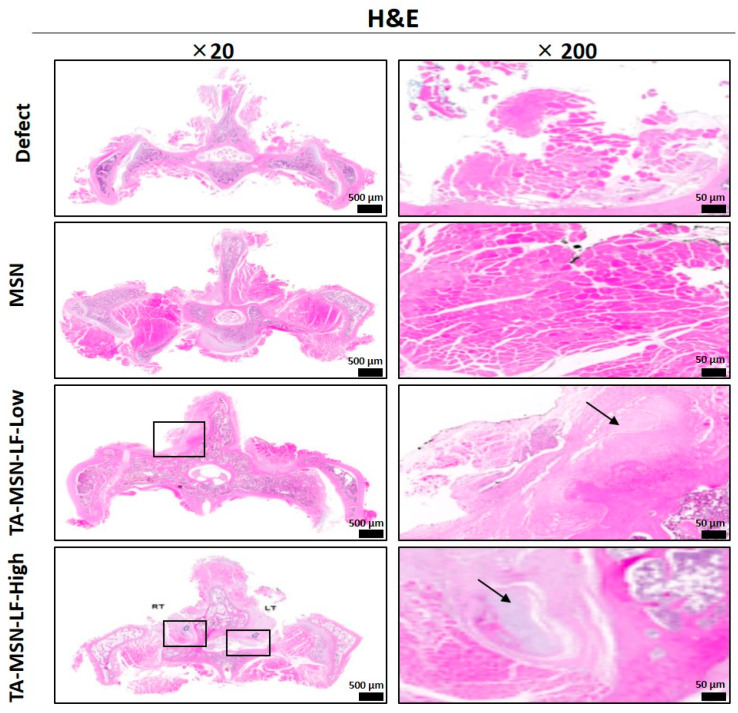
Histological analysis of the spinal fusion animal model. Histological sections were stained with H&E and boxes/black arrows indicate new bone formation. Scale bar = 500/50 µm.

**Figure 3 ijms-24-15782-f003:**
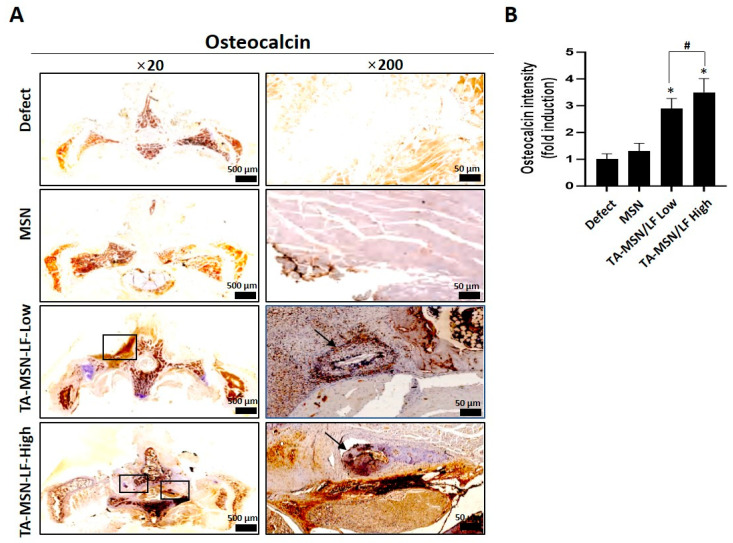
(**A**) Immunohistochemistry for osteocalcin; (**B**) Osteocalcin intensity. The data were expressed as mean ± SEM in each bar graph. * *p* < 0.05 compared with the defect group. ^#^
*p* < 0.05 between the two groups. The boxes/black arrows indicate new bone formation. Scale bar = 500/50 µm.

**Figure 4 ijms-24-15782-f004:**
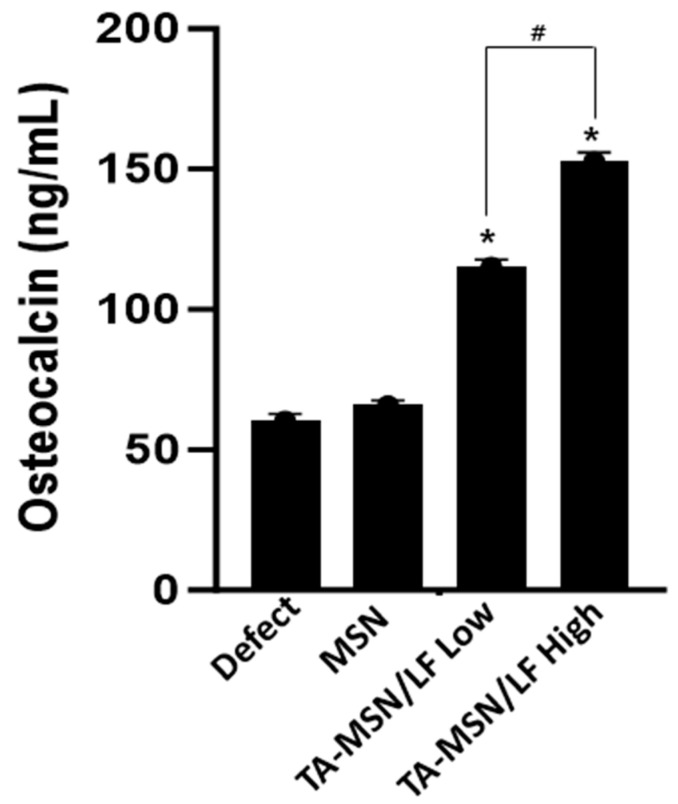
Serum osteocalcin by ELISA. The data was expressed as mean ± SEM in each bar graph. * *p* < 0.05 compared with the defect group. ^#^
*p* < 0.05 between the two groups.

**Figure 5 ijms-24-15782-f005:**
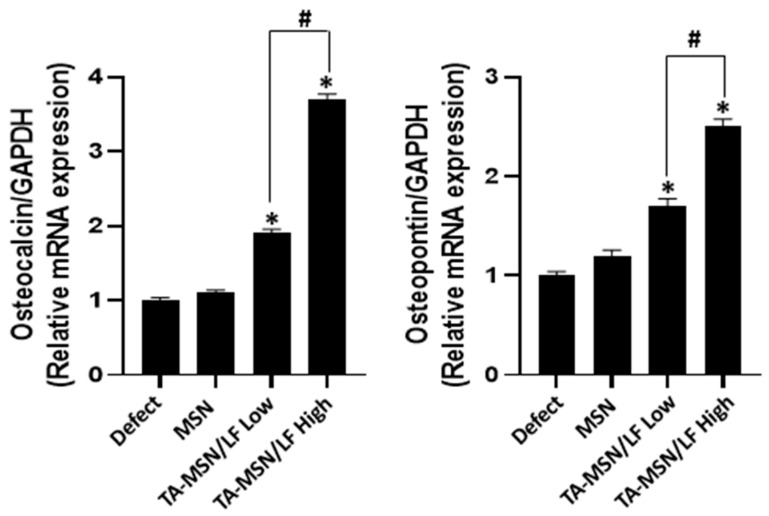
The mRNA expression of osteocalcin and osteopontin. The relative mRNA expression of each target was normalized to the GAPDH. Values are the means ± SEM in each bar graph. * *p* < 0.05 compared with the defect group. ^#^
*p* < 0.05 between the two groups.

**Figure 6 ijms-24-15782-f006:**
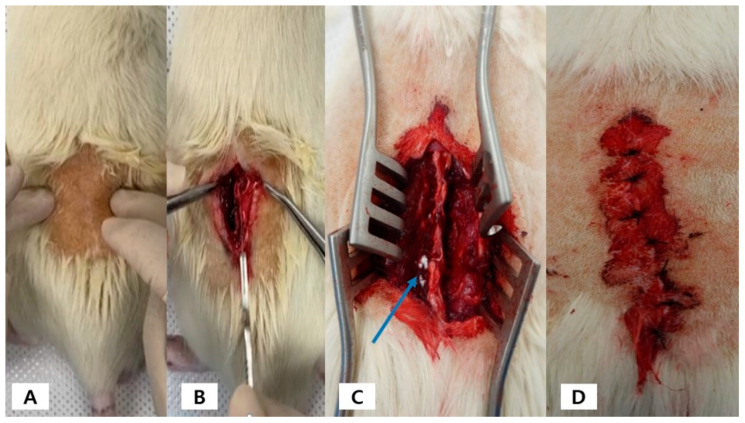
(**A**) After anesthetizing the rat, the hair of the experimental area is shaved; (**B**) The skin is incised, and the muscles are dissected; (**C**) Materials were placed on the decorated area (blue arrow); (**D**) The fascia and skin were sutured.

**Table 1 ijms-24-15782-t001:** Qualitative fusion assessment by manual palpation at 8 weeks after fusion.

Characteristics	MSN Group (*n* = 8)	TA-MSN-LF Low Group (*n* = 8)	TA-MSN-LF High Group (*n* = 8)	*p*
Fusion grade				
1	2	0	0	
2	6	3	1	
3	0	5	7	<0.001 **
Degree of firmness				
1	5	0	0	
2	3	4	0	
3	0	4	8	<0.001 **
Volume of bridging bone				
1	2	0	0	
2	5	2	0	
3	1	6	8	<0.001 **

** *p* value < 0.01.

**Table 2 ijms-24-15782-t002:** Quantitative results of micro-CT findings at 8 weeks after fusion.

	Defect Group (*n* = 6)	MSN Group (*n* = 8)	TA-MSN-LF Low Group (*n* = 8)	TA-MSN-LF High Group (*n* = 8)	*p*
Bone volume (mm^3^)	84.761 ± 8.507	99.697 ± 14.234	114.701 ± 15.703	121.335 ± 4.204	<0.05 *
Trabecular thickness (mm)	0.485 ± 0.058	0.400 ± 0.029	0.416 ± 0.046	0.412 ± 0.047	0.348
Trabecular number (1/mm)	0.152 ± 0.022	0.179 ± 0.030	0.211 ± 0.015	0.212 ± 0.020	<0.05 *
Trabecular separation (mm)	4.500 ± 0.343	4.409 ± 0.246	4.277 ± 0.238	4.307 ± 0.425	0.637

* *p* value < 0.05.

**Table 3 ijms-24-15782-t003:** Semiquantitative histological evaluation of the frequency of blood vessels.

	Defect Group (*n* = 6)	MSN Group (*n* = 8)	TA-MSN-LF Low Group (*n* = 8)	TA-MSN-LF High Group (*n* = 8)	*p*
Frequency of blood vessels	1.00 ± 0.00	1.37 ± 0.50	1.51 ± 0.50	1.96 ± 0.50	<0.05 *

* *p* value < 0.05.

## Data Availability

Not applicable.

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
