# Peer review of "Lactoferrin-Anchored Tannylated Mesoporous Silica Nanomaterials-Induced Bone Fusion in a Rat Model of Lumbar Spinal Fusion"

_ijms, 2023, doi:10.3390/ijms242115782_

Round 1
Reviewer 1 Report
Comments and Suggestions for Authors
Although the subject of the manuscript” Lactoferrin-anchored tannylated mesoporous silica nanomaterials induced bone fusion in a rat model of lumbar spinal fusion” addressed by the authors is interesting, the presentation it is not convincing.
-Introduction: the aim of the study presented at the end of Section is too short presented;
-Results: brief presentation, with very little explanations/comments; Why the authors have chosen low and high lacoferrin concentration? On what basis? And more, which formulation is better to be used, the low or high? A comparison between the two formulations is not presented in Fig. 3B, 4 and 5, if any (*p relevant?)
- Discussion: comments related to the biological significance of the results are missing. Even their article mentioned in the text Noh et al., 2020 is not inserted in the References!!
- A final English spelling is necessary.
Finally, as authors have mentioned, there are several limitations very important for the study, such as the effect of Lactoferrin formulation vs other graft materials or the degree of healing over time.
Overall, I find this paper premature and unclear in its connection to therapeutics needs. The authors might consider controls to be used as a comparison point of advantages their system might provide.
At this stage, the reviewer does not recommend the publication of this article on International Journal of Molecular Sciences.
Comments on the Quality of English LanguageAlthough the subject of the manuscript” Lactoferrin-anchored tannylated mesoporous silica nanomaterials induced bone fusion in a rat model of lumbar spinal fusion” addressed by the authors is interesting, the presentation it is not convincing.
-Introduction: the aim of the study presented at the end of Section is too short presented;
-Results: brief presentation, with very little explanations/comments; Why the authors have chosen low and high lacoferrin concentration? On what basis? And more, which formulation is better to be used, the low or high? A comparison between the two formulations is not presented in Fig. 3B, 4 and 5, if any (*p relevant?)
- Discussion: comments related to the biological significance of the results are missing. Even their article mentioned in the text Noh et al., 2020 is not inserted in the References!!
- A final English spelling is necessary.
Finally, as authors have mentioned, there are several limitations very important for the study, such as the effect of Lactoferrin formulation vs other graft materials or the degree of healing over time.
Overall, I find this paper premature and unclear in its connection to therapeutics needs. The authors might consider controls to be used as a comparison point of advantages their system might provide.
At this stage, the reviewer does not recommend the publication of this article on International Journal of Molecular Sciences.
Author Response
Reviewer #1:
Although the subject of the manuscript” Lactoferrin-anchored tannylated mesoporous silica nanomaterials induced bone fusion in a rat model of lumbar spinal fusion” addressed by the authors is interesting, the presentation it is not convincing.
-Introduction: the aim of the study presented at the end of Section is too short presented;
Response: Thank you for your comment. We have supplemented the introduction section, accordingly. The introduction section was modified as follows:
“1. Introduction
Spinal fusion is an increasingly common procedure used to treat various pathologies arising from trauma, degenerative diseases, infections, and tumors. [1] However, pseudarthrosis can occur because of a failure of bone fusion, resulting in pain, instability, and disability. [2] Failure of bone bridging increases the risk of device failure, with sedimentation or screw loosening. [3] Pseudarthrosis rates range from 5% to 34%. [4] Successful bone fusion after spinal fusion is challenging, particularly in elderly patients with osteoporosis, due to low bone strength and poor bone quality. Over the past decade, a wide range of treatment modalities to prevent fusion failure or pseudarthrosis have been explored, with research focusing on the identification of new substances that aid fusion.
Employing autograft iliac crest bone is the clinical "gold standard" in fusion procedures. However, these autografts have several disadvantages. Collecting a consistent volume of the iliac crest bone may be difficult, and the harvesting procedure can be painful and may be associated with hematomas, infections, fractures at the donor location, and prolonged surgical time. Furthermore, there are limitations in the use of bone autografts with regards to patients with osteoporosis or metabolic diseases.
Except for autologous bone graft material, spine surgeons have used various methods, including allogeneic bone graft materials (e.g. demineralized bone allograft), synthetic bone graft materials (e.g. tricalcium phosphate, hydroxyapatite, bioglass), and bone morphogenetic protein-2 (BMP2) to achieve bone fusion. [5] BMP2 was approved in 2002 and is currently used for spinal fusion. Although BMP2 is effective in osteogenic differentiation and osteogenesis, delivery of BMP2 alone is not effective in supporting osteogenesis due to its short half-life. BMP2 diffuses rapidly throughout body fluids and is cleared rapidly. [6] Therefore, higher than physiological doses of BMP2 have been used clinically to promote bone formation. However, high doses of BMP2 are associated with serious complications such as soft tissue inflammation and heterotopic ossification. [7]
Lactoferrin (LF) is an 80 kDa non-heme iron-binding protein produced primarily by exocrine epithelial cells and expressed in most tissues that plays an important role in the innate immune system of mammals. [8] LF is a potent antiviral, anti-inflammatory, and antibacterial agent found in cow and human colostrum. [9] Moreover, LF acts as an osteogenic growth factor. A previous report showed that treatment with LF not only increased osteoblast proliferation but also promoted osteoblast differentiation. [8] Previous studies examined the oral administration or local injection of LF. [10-12] Yanagisawa et al. found that oral administration of LF suppressed the progression of rheumatoid arthritis in a mouse model. [10] Guo et al. observed that oral administration of LF to ovariectomized rats resulted in preserved bone mass but also improved bone microarchitecture. [11] Furthermore, Cornish et al. reported that local injection of LF over the hemicalvaria increases bone growth in adult male mice. [12]
A previous study examined LF-anchored tannylated mesoporous silica nanomaterials (TA-MSN-LF). [13] LF is a promising protein for osteogenic differentiation and tendon healing; however, without the assistance of a delivery medium, the effect of the protein drug was not as effective as expected because of its short half-life in the blood. [14] Therefore, recent research has focused on the use of MSN and TA. MSN are attracting considerable attention for their potential in biomedical applications because they have excellent biocompatibility and biodegradability compared to other inorganic nanomaterials. [15] Further advantages of MSN from a bone tissue engineering perspective are the large surface area, large pore size, and easy surface modification. [16] TA functions as both an organic and inorganic sample due to the presence of hydroxyl and galloyl groups. [17] TA is known to interact directly with several types of biopolymers (DNA, gelatin, collagen, albumin, chitosan, thrombin, enzymes) because of its electrostatic interaction, hydrogen bonding, and hydrophobic interaction abilities. [18] The osteo-differentiation ability of TA-MSN-LF has been demonstrated in vitro. [13] This study aimed to investigate whether TA-MSN-LF functions as a bone fusion material in a rat model of lumbar spinal fusion. This is the first study to demonstrate the effectiveness of lactoferrin in an animal model of lumbar spinal fusion.”
-Results: brief presentation, with very little explanations/comments;
Response: Thank you for your comment. We have supplemented the results section, accordingly. The results section was modified as follows:
“2. Results
Of the 33 rats, three died during surgery due to side effects of general anesthesia, and 30 survived without any complications. The behavior of the animals showed no specific changes suggesting side effects of the chemicals or local complications such as nerve irritation. Finally, the lumbar fusion models used in the analysis were as follows: TA-MSN-LF high group, 8; TA-MSN-LF low group, 8; MSN group, 8; and Defect group, 6.
2.1. Manual palpation
Table 1 shows the qualitative fusion assessment by manual palpation at 8 weeks after fusion. Qualitative parameters such as degree of fusion, degree of hardness, and volume of the bridged bone were significantly different between each group (p<0.001). Additionally, the manual qualitative fusion scores were significantly different between each group (p<0.001). All parameters derived from manual palpation tended to show better results in the TA-MSN-LF high group than in the other groups. These results demonstrated that promotion of bone fusion was superior in the TA-MSN-LF high group.
2.2. Micro-CT
Table 2 shows the quantitative results of the micro-computed tomography (CT) findings 8 weeks after fusion. Micro-CT of the TA-MSN-LF high and low groups revealed new bone formation in the decortication area (Figure 1). Additionally, quantitative fusion bone volume and trabecular bone number were significantly different among the groups (p<0.05). Bone volume and trabecular number in the TA-MSN-LF high group were higher than those in the other groups. However, there were no significant differences in trabecular bone thickness and trabecular separation among the four groups. These results demonstrated that promotion of bone fusion was superior in the TA-MSN-LF high group.
2.3. Histologic results
Hematoxylin eosin (H&E) and osteocalcin (OCN) staining were performed to evaluate new bone formation 8 weeks after the procedure. As shown in Figure 2, histological sections were stained with H&E and the boxes indicate new bone formation in the TA-MSN-LF high and low groups. Figure 3A shows staining for OCN, a bone formation marker, in each group. Staining was best performed in the TA-MSN-LF-high group. Figure 3B quantifies this and compares the osteocalcin levels. The TA-MSN-LF high and low groups had significantly higher OCN intensity than that in the other groups (p<0.05). The OCN intensity of the TA-MSN-LF-high group was significantly higher than that in the TA-MSN-LF-low group (p<0.05). Moreover, the blood vessel frequency in the TA-MSN-LF high group was significantly higher than that in the other groups (p<0.05, Table 3) The histologic results indicate that bone fusion and angiogenesis were most effective in the TA-MSN-LF high group.
2.4. Bone turnover marker
The serum OCN concentration was significantly higher in the TA-MSN-LF high and low groups than that in the other groups (p<0.05, Figure 4). The serum OCN concentration of the TA-MSN-LF-high group was significantly higher than that in the TA-MSN-LF-low group (p<0.05). The serum bone turnover marker results show that bone fusion ability was the highest in the TA-MSN-LF high group.
2.5. Quantification of osteo-differentiation specific genes
The mRNA expression levels of OCN and osteopontin (OPN) were significantly higher in the TA-MSN-LF high and low groups than those in the other groups (p<0.05, Figure 5). The mRNA expression level of OCN and OPN in the TA-MSN-LF-high group was significantly higher than that in the TA-MSN-LF-low group (p<0.05). This finding may indicate that the TA-MSN-LF-high may promote and accelerate bone formation.”
Why the authors have chosen low and high lacoferrin concentration? On what basis? And more, which formulation is better to be used, the low or high?
Response: Thank you for your comment. The use of TA-MSN-LF low (1 µg/mL) and TA-MSN-LF high (100 µg/mL) in our study was determined based on previous studies using lactoferrin. [31] Chang et al. evaluated the osteo-differentiation of adipose-derived stem cells by lactoferrin. The doses of lactoferrin investigated in the study were 10 µg/mL, 20 µg/mL, 50 µg/mL, 100 µg/mL, and 500 µg/mL. The authors reported that 100 µg/mL was the most effective dose. Therefore, in our study, we compared 1 µg/mL and 100 µg/mL of LF. TA-MSN-LF high (100 µg/mL) was superior to TA-MSN-LF low (1 µg/mL) across all results. Finally, the bone fusion ability of TA-MSN-LF high (100 µg/mL) was higher than that of TA-MSN-LF low (1 µg/mL). We have revised the Materials and Methods section as follows:
“4.2. Materials
Mesoporous silica nanomaterials (MSNs) (Sigma-Aldrich) were modified with tannic acid (TA) (Sigma-Aldrich) to immobilize human LF (Sigma-Aldrich, St. Louis, MO, USA). First, 10 mg of MSN were added to a phosphate buffered saline (PBS) solution (pH, 7.4) containing dissolved TA (concentration, 50 μg/mL) and the mixture was gently shaken overnight at room temperature (RT). MSNs containing TA were then washed twice with distilled water (DW) and lyophilized for two days. The TA load on MSN was confirmed by analyzing the amount of residual TA in the PBS solution. Hereafter, the MSNs with TA are referred to as TA-MSNs.
To immobilize LF (concentration, 1 or 100 μg/mL) on the MSN surface, TA-MSN (concentration, 10 µg/mL) and LF (concentration, 1 or 100 μg/mL) were added to the PBS solution and then incubated for 24 h. Next, all samples were washed three times with DW at 3000 rpm and 4 °C for 10 min using a Smart R17 Centrifuge (Hanil Science Industrial, Incheon, Korea). The samples were freeze-dried for two days.
To assess the LF load, the supernatant was collected after the immobilization of LF on TA-MSN and analyzed using the Pierce Bicinchoninic Acid (BCA) Protein Assay Kit (Thermo Fisher Scientific, Rockford, IL, USA), according to the manufacturer's protocol. The LF (1 µg/mL) anchor-type TA-MSN and LF (100 µg/mL) anchor-type TA-MSN are hereafter referred to as TA-MSN-LF low and TA-MSN-LF high, respectively. The LF dose, based on the addition of 3mg of TA-MSN-LF, was calculated as 0.03μg/rat (1μg/mL TA-MSN-LF) and 2.75 μg/rat (100μg/mL TA-MSN-LF). The use of TA-MSN-LF low (1 µg/mL) and TA-MSN-LF high (100 µg/mL) in our study was based on previous studies using LF. [31] Chang et al. evaluated the osteo-differentiation of adipose-derived stem cells by LF dose (10 µg/mL, 20 µg/mL, 50 µg/mL, 100 µg/mL, and 500 µg/mL) and concluded that100 µg/mL was the most effective dose. Therefore, in the present study, we compared 1 µg/mL and 100 µg/mL doses.”
A comparison between the two formulations is not presented in Fig. 3B, 4 and 5, if any (*p relevant?)
Response: Thank you for your comment. We have set out the comparison between TA-MSN-LF high and TA-MSN-LF low in Fig. 3B, 4 and 5.
- Discussion: comments related to the biological significance of the results are missing. Even their article mentioned in the text Noh et al., 2020 is not inserted in the References!!
Response: Thank you for your comment. We have revised the Discussion section and amended the references, as follows:
“3. Discussion
Although the development of new substances for bone-tissue regeneration remains challenging, numerous biomaterials have been proposed for this purpose. LF is an osteoinductive factor that can promote the osteogenic differentiation of various cells, such as osteoblasts, C2C12 cells, and adipose-derived stem cells (ADSC). [19] In a previous study, we developed a new type of TA-MSN-LF to release LF from nanoparticles over a long period of time and impart osteoconductive effects to the silica nanoparticles. [13] In this study, this material was placed into rats to determine its effectiveness as a bone fusion material.
LF is a widely distributed glycoprotein in mammals; human tears contain approximately 1.13 g/L of LF and human colostrum contains more than 5 g/L. [20,21] LF is a crucial anabolic bone growth factor and can function as an effector molecule for bone remodeling. [22] In vitro experiments have shown that local distribution of LF strongly increased the proliferation and differentiation of osteoblasts from mice to humans. [23,24] Other studies have demonstrated that the oral administration of LF not only improves bone formation and sclerosis in distraction osteogenic animal models, but also improves bone mineral density (BMD) and biomechanical strength in ovariectomized animal models. [25,26]
To construct the TA-MSN-LF, LF was immobilized on the MSN surface coated with TA. TA is a representative molecule with strong affinity for various proteins, such as proline-rich ones. [27] Each MSN group was observed by transmission electron microscope (TEM), and dynamic light scattering (DLS) was performed in the range of 260 –300 nm. We also used X-ray photoelectron spectroscopy (XPS) to investigate the surface chemical composition of MSNs containing TA and/or LF, and compared this with the composition of pure MSNs. TA-MSN-LF exhibited a sustained release of LF for up to 28 days; these results demonstrated that TA-MSN-LF nanoparticles can achieve long-term delivery of LF. Park et al. investigated whether local bone regeneration occurred in a rat calvarial defect model using LF and poloxamer gels, [28] which were used as the LF carriers. However, LF was released for up to three days under these conditions. In conclusion, the LF/poloxamer material increased cell viability and was less likely to induce immune responses; however, the formulation failed to enhance bone regeneration in vivo. Conversely, the role of the TA-MSN as a transmitter was significant in our study.
In this study, we evaluated whether TA-MSN-LF could induce the formation of bone tissue and spinal regeneration in vivo as an alternative to autologous bone. In a rat model, TA-MSN-LF significantly increased new bone tissue formation and vertebral bone regeneration 8 weeks after a spinal fusion procedure; additionally, the bone volume and trabecular number were significantly higher in the high-dose TA-MSN-LF group (<0.05). Guo et al. analyzed the effects of oral LF on bone mass and microarchitecture in a rat model of osteoporosis. [11] The LF dose was divided into three groups: 0.85 mg/kg body weight, 8.5 mg/kg body weight, and 85 mg/kg body weight. Micro-CT analysis revealed that bone volume was the highest in the 85 mg/kg body weight group. Koca et al. investigated whether LF was beneficial for autograft healing in peri-implant bone in a rat model; [29] the percentage of bone volume (bone volume/total volume) in the micro-CT analysis was significantly higher in the defect filled with autograft and LF (100 μg/mL) group that in the autograft only group. Thus, TA-MSN-LF conclusively induces bone tissue formation and spinal regeneration.
Park et al. analyzed the effects of LF in a rat model of calvarial defects. [28] Histologic analysis evaluating the osteogenic effects of LF revealed that bone defects were reduced in the group injected with LF compared to those in the control group (p<0.05). Guo et al. reported that a group treated with LF presented higher serum OCN levels than those in the control group. [11] The authors examined osteoblast-like cell metabolic activity at 24 h, which increased in the group that consumed LF. In the present study, OCN was stained and quantified. In the TA-MSN-LF high and low groups, OCN staining was good, and the OCN intensity was significantly higher than that in the other groups. Additionally, the mRNA expression levels of OCN and OPN were significantly higher in the high AND low TA-MSN-LF groups than those in the other groups (p<0.05).
As a biologically active molecule, LF promotes proliferation and differentiation of various cells. [19] It also inhibits osteoclast formation by reducing the number of osteoclasts that can actively resorb bone. [30] These positive effects of LF support the use of LF in bone tissue regeneration. However, due to the low bioavailability of LF in vivo, nanomaterial-based strategies have been developed to improve the biological activity of LF. This in vitro study reported that TA-MSN-LF promotes osteogenesis and angiogenesis. Although this study did not compare TA-MSN-LF with other materials, Chang et al. reported that 100ug of LF produced more substantial osteoinductive effects than BMP-2 [31], concluding that LF could replace BMP-2. [31] Considering the importance of bone fusion ability, the TA-MSN-LF high (100 μg/mL) was excellent in inducing differentiation and promoting proliferation in bone tissue engineering.”
- A final English spelling is necessary.
Response: Thank you for your comment. As suggested, the manuscript has been reviewed for English spelling and language.
Finally, as authors have mentioned, there are several limitations very important for the study, such as the effect of Lactoferrin formulation vs other graft materials or the degree of healing over time.
Response: Thank you for your comment. As noted in the limitations, this study did not compare different substances and did not analyze effects over time. However, Chang et al. compared BMP-2 and LF and reported that LF 100 had more osteoinductive effects than BMP-2. [31] We will conduct further research by referring to this study.
Overall, I find this paper premature and unclear in its connection to therapeutics needs. The authors might consider controls to be used as a comparison point of advantages their system might provide.
Response: Thank you for your comment. This is the first in vivo study using lactoferrin for bone fusion, and there are many shortcomings in its immediate clinical application. However, Chang et al. analyzed the mechanisms underlying BMP-2, which is currently used as a bone fusion material. [31] As shown in this paper, lactoferrin has many advantages as a bone fusion material. In our next study, we will address the shortcomings mentioned in the limitations section.

Reviewer 2 Report
Comments and Suggestions for Authors
I find the work interesting but there are things that are not very clear.
It is not understood how an in situ treatment related to an intervertebral fusion can affect the levels of circulating bone differentiation. Furthermore, it is not clear from the presentation where they obtain the samples to do RTPCR.
materials and methods should be better explained according to what is indicated below
Tanic acid was measured by analyzing the amount of residual TA in the PBS solution? How they do it?
. Bone turnover marker The quantitative osteogenic ability of each group to induce osteoblast activity was compared by evaluating serum OCN levels using an OCN ELISA kit (Novus, CA, USA). All standards and samples were analyzed using an iMark microplate reader (Bio-Rad, CA, 274 USA) at a wavelength of 450 nm}
Really in a local supplementation they observed serum differences? At what time post-surgery the the authors study the serum samples?
Quantification of osteo-differentiation specific genes To further confirm the osteogenic capacity of MSNs with or without LF, the mRNA levels of osteogenic differentiation-specific genetic markers such as OCN and OPN were 278 examined using real-time polymerase chain reaction (PCR).
From what samples they obtain the mRNA in order to study the osteogenic differentiation-specidifec genetic matkers? At what post-surgical time they made these study?
How the authors euthanized the animals
Author Response
Reviewer #2:
I find the work interesting but there are things that are not very clear.
It is not understood how an in situ treatment related to an intervertebral fusion can affect the levels of circulating bone differentiation.
Response: Thank you for your comment. Chang et al., whom I cited as a reference, clearly explained the mechanism by which lactoferrin prompts osteogenesis. To explore the mechanism by which LF promotes osteogenic differentiation of adipose-derived stem cells, RNA sequencing was performed on a population of adipose-derived stem cells treated with 100 μg/mL LF. The authors used the KEGG and Reactome databases. In the KEGG databases, the PI3K/AKT and MAPK pathways and the interaction between cytokines and their receptors were upregulated. Reactome enrichment revealed that extracellular matrix organization pathway up-regulation, and regulation of insulin-like growth factor (IGF) transport and uptake by insulin-like growth factor binding protein (IGFBPS) were upregulated. Through this mechanism, osteogenesis is prompted when TA-MSN-LF is implanted to a spinal bone fusion model.
Furthermore, it is not clear from the presentation where they obtain the samples to do RTPCR.
Response: Thank you for your comment. Eight weeks after surgery, blood was drawn from the tail vein under zoletyl-xylazine anesthesia. We have revised the Material and Methods section as follows:
“4.8. Quantification of osteo-differentiation specific genes
To further confirm the osteogenic capacity of MSNs with or without LF, the mRNA levels of osteogenic differentiation-specific genetic markers such as OCN and OPN were examined using real-time polymerase chain reaction (PCR). Eight weeks after surgery, blood was collected from the tail vein under zoletyl-xylazine anesthesia. Total RNA was purified using A TRIzol reagent and cDNA was prepared from total RNA (1 µg) using a PrimeScriptTM 1st Strand cDNA Synthesis Kit (Takara Bio Inc., Ostu, Japan). PCR amplification and real-time PCR were performed using an ABI7300 Real-Time Thermal Cycler (Applied Biosystems, Foster City, CA, USA). OCN and OPN genes were normalized to glyceraldehyde 3-phophate dehydrogenase (GAPDH). All experiments were repeated three times at each time point. Expression levels were calculated using the 2-△△CT method.”
materials and methods should be better explained according to what is indicated below
Tanic acid was measured by analyzing the amount of residual TA in the PBS solution? How they do it?
Response: Thank you for raising this question. The TA coating on the MSN surface was quantified by a biocinchoninic acid (BCA) assay as follow: the harvested supernatant PBS was mixed with a BCA working solution, followed by incubation for 1 h at 37°C. After incubation, the absorbance of the resulting solution was monitored using a Multimode Reader (VarioskanTM, Thermo Scientific, Waltham, MA, USA). The TA amount on the MSN surface was determined based on the standard curve for TA (Y = 0.0075x + 0.074, R2 = 0.99; TA concentration range of standard curve: 0 to 500 mg/mL).
Bone turnover marker
The quantitative osteogenic ability of each group to induce osteoblast activity was compared by evaluating serum OCN levels using an OCN ELISA kit (Novus, CA, USA). All standards and samples were analyzed using an iMark microplate reader (Bio-Rad, CA, 274 USA) at a wavelength of 450 nm}
Really in a local supplementation they observed serum differences? At what time post-surgery the the authors study the serum samples?
Response: Thank you for raising this question. Serum differences were identified. Eight weeks after surgery, blood was drawn from the tail vein under zoletyl-xylazine anesthesia. The Materials and Methods section were revised as follows:
“4.7. Bone turnover marker
The quantitative osteogenic ability of each group to induce osteoblast activity was compared by evaluating serum OCN levels using an OCN ELISA kit (Novus, CA, USA).
Eight weeks after surgery, blood was collected from the tail vein under zoletyl-xylazine anesthesia. All standards and samples were analyzed using an iMark microplate reader (Bio-Rad, CA, USA) at a wavelength of 450 nm.”
Quantification of osteo-differentiation specific genes
To further confirm the osteogenic capacity of MSNs with or without LF, the mRNA levels of osteogenic differentiation-specific genetic markers such as OCN and OPN were 278 examined using real-time polymerase chain reaction (PCR).
From what samples they obtain the mRNA in order to study the osteogenic differentiation-specidifec genetic matkers? At what post-surgical time they made these study?
Response: Thank you for raising this question. Eight weeks after surgery, blood was drawn from the tail vein under zoletyl-xylazine anesthesia. The Materials and Methods section were revised as follows:
“4.8. Quantification of osteo-differentiation specific genes
To further confirm the osteogenic capacity of MSNs with or without LF, the mRNA levels of osteogenic differentiation-specific genetic markers such as OCN and OPN were examined using real-time polymerase chain reaction (PCR). Eight weeks after surgery, blood was collected from the tail vein under zoletyl-xylazine anesthesia. Total RNA was purified using A TRIzol reagent and cDNA was prepared from total RNA (1 µg) using a PrimeScriptTM 1st Strand cDNA Synthesis Kit (Takara Bio Inc., Ostu, Japan). PCR amplification and real-time PCR were performed using an ABI7300 Real-Time Thermal Cycler (Applied Biosystems, Foster City, CA, USA). OCN and OPN genes were normalized to glyceraldehyde 3-phophate dehydrogenase (GAPDH). All experiments were repeated three times at each time point. Expression levels were calculated using the 2-△△CT method.”
How the authors euthanized the animals
Response: Thank you for your comment. All rats were euthanized using anesthesia gas. The Materials and Methods section were modified as follows:
“4.3. Experimental design and surgical procedure
A single-level bilateral lumbar posterolateral fusion was performed, and thirty host rats were divided into four experimental groups: [A] LF (100 µg) anchor-type TA-MSN (n = 8), [B] LF (1 µg) anchor-type TA-MSN (n = 8), [C] MSN (n = 8), and [D] Defect (n=6). After anesthesia with a mixed solution of zoletil-xylazine (zoletil at 20 mg/kg and xylazine at 10 mg/kg), the hair on the surgical site was shaved. The vertebral levels from L4 to L5 were identified by palpation and anatomical landmarks. A dorsal midline skin incision was made at the center of the L4-L5 spinous processes, and the edges of the skin were retracted using a self-holding aspirator. An intermuscular plane was established between the multifidus and longissimus muscles, thereby exposing the transverse process from L4 to L5. Decortication of the transverse processes and external segments/intervertebral joints was performed using an electric bar. After decorticating the transverse processes on both sides of L4 and L5, 3 mg of each material was implanted on each side of the fusion bed space (L4-L5). After suturing the muscle and skin layer-by-layer, cefazolin (100 mg/kg) was injected. The rats were euthanized eight weeks after the experiment with an anesthetic gas. Figure 6 illustrates the experimental animal process.”

Round 2
Reviewer 1 Report
Comments and Suggestions for Authors
The manuscript entitle “Lactoferrin-anchored tannylated mesoporous silica nanomaterials induced bone fusion in a rat model of lumbar spinal fusion” was revised and the version 2 revealed an improved form.
However, the authors still need to clarify why they used in the experiments the formulation with low concentration of Lactoferrin (1μg/ml). This formulation was not used and characterize by Noh et al 2020. Also Chang et al., 2023 have investigated the osteogenic effect of Lactoferrin for concentrations ranging from 10 to 500μg/ml.
In conclusion the paper can be published only after the above mentioned clarification.
Comments on the Quality of English Language
Only minor English style improvement.
Author Response
Reviewer #1:
The manuscript entitle “Lactoferrin-anchored tannylated mesoporous silica nanomaterials induced bone fusion in a rat model of lumbar spinal fusion” was revised and the version 2 revealed an improved form.
Response: Thank you for your comment.
However, the authors still need to clarify why they used in the experiments the formulation with low concentration of Lactoferrin (1μg/ml). This formulation was not used and characterize by Noh et al 2020. Also Chang et al., 2023 have investigated the osteogenic effect of Lactoferrin for concentrations ranging from 10 to 500μg/ml. In conclusion the paper can be published only after the above mentioned clarification.
Response: Thank you for your comment. The use of TA-MSN-LF low (1 µg/mL) and TA-MSN-LF high (100 µg/mL) in our study was determined based on previous studies using lactoferrin. [31-33] Chang et al. evaluated the osteo-differentiation of adipose-derived stem cells by lactoferrin. The doses of lactoferrin investigated in the study were 10 µg/mL, 20 µg/mL, 50 µg/mL, 100 µg/mL, and 500 µg/mL. [31] Li et al. studied the osteoblast differentiation ability of lactoferrin. [32] In this study, lactoferrin doses of 1μg, 10μg, 100μg, and 1000μg were studied. [32] Among these, 100μg was reported to be the most effective. [32] And in a study by Zhang et al., doses of lactoferrin of 1μg, 10μg, and 100μg were studied and it was reported that 100μg was the most effective. [33] We have revised the Materials and Methods section and discussion section as follows:
“Results
Park et al. analyzed the effects of LF in a rat model of calvarial defects. [28] Histologic analysis evaluating the osteogenic effects of LF revealed that bone defects were reduced in the group injected with LF compared to those in the control group (p<0.05). Guo et al. reported that a group treated with LF presented higher serum OCN levels than those in the control group. [11] The authors examined osteoblast-like cell metabolic activity at 24 h, which increased in the group that consumed LF. In the present study, OCN was stained and quantified. In the TA-MSN-LF high and low groups, OCN staining was good, and the OCN intensity was significantly higher than that in the other groups. Additionally, the mRNA expression levels of OCN and OPN were significantly higher in the high AND low TA-MSN-LF groups than those in the other groups (p<0.05). As a biologically active molecule, LF promotes proliferation and differentiation of various cells. [19] It also inhibits osteoclast formation by reducing the number of osteoclasts that can actively resorb bone. [30] These positive effects of LF support the use of LF in bone tissue regeneration very interesting. However, due to the low bioavailability of LF in vivo, nanomaterial-based strategies have been developed to improve the biological activity of LF. This in vitro study reported that TA-MSN-LF promotes osteogenesis and angiogenesis. Although this study did not compare TA-MSN-LF with other materials, Chang et al. reported that 100ug of LF produced more substantial osteoinductive effects than BMP-2 [31], concluding that LF could replace BMP-2. [31] Li et al. studied the osteoblast differentiation ability of lactoferrin and reported that 100ug was most effective. [32] And Zhang et al.'s study also reported that 100ug of lactoferrin was most effective in proliferative activity of the osteoblast cells. [33] Considering the importance of bone fusion ability, the TA-MSN-LF high (100 μg/mL) was excellent in inducing differentiation and promoting proliferation in bone tissue engineering in our study.”
“4.2. Materials
Mesoporous silica nanomaterials (MSNs) (Sigma-Aldrich) were modified with tannic acid (TA) (Sigma-Aldrich) to immobilize human LF (Sigma-Aldrich, St. Louis, MO, USA). First, 10 mg of MSN were added to a phosphate buffered saline (PBS) solution (pH, 7.4) containing dissolved TA (concentration, 50 μg/mL) and the mixture was gently shaken overnight at room temperature (RT). MSNs containing TA were then washed twice with distilled water (DW) and lyophilized for two days. The TA load on MSN was confirmed by analyzing the amount of residual TA in the PBS solution. Hereafter, the MSNs with TA are referred to as TA-MSNs.
To immobilize LF (concentration, 1 or 100 μg/mL) on the MSN surface, TA-MSN (concentration, 10 µg/mL) and LF (concentration, 1 or 100 μg/mL) were added to the PBS solution and then incubated for 24 h. Next, all samples were washed three times with DW at 3000 rpm and 4 °C for 10 min using a Smart R17 Centrifuge (Hanil Science Industrial, Incheon, Korea). The samples were freeze-dried for two days.
To assess the LF load, the supernatant was collected after the immobilization of LF on TA-MSN and analyzed using the Pierce Bicinchoninic Acid (BCA) Protein Assay Kit (Thermo Fisher Scientific, Rockford, IL, USA), according to the manufacturer's protocol. The LF (1 µg/mL) anchor-type TA-MSN and LF (100 µg/mL) anchor-type TA-MSN are hereafter referred to as TA-MSN-LF low and TA-MSN-LF high, respectively. 3mg was used in the experiment, and the real LF dose was calculated when 3mg of TA-MSN-LF was added, which is 0.03μg/rat (1μg/mL TA-MSN-LF) and 2.75 μg/rat (100μg/mL TA-MSN-LF). The use of TA-MSN-LF low (1 µg/mL) and TA-MSN-LF high (100 µg/mL) in our study was based on previous studies using LF. [31-33] Chang et al. evaluated the osteo-differentiation of adipose-derived stem cells LF dose (10 µg/mL, 20 µg/mL, 50 µg/mL, 100 µg/mL, and 500 µg/mL) and concluded that 100 µg/mL was the most effective dose. [31] In Li et al.’s study, lactoferrin doses of 1μg, 10μg, 100μg, and 1000μg were studied. And in a study by Zhang et al., doses of lactoferrin of 1μg, 10μg, and 100μg were studied. Therefore, in the present study, we compared 1 µg/mL and 100 µg/mL doses.”

Reviewer 2 Report
Comments and Suggestions for Authors
the paper could be published, but the reviewer did not understand yet hoy osteocalcin serum level could be modified in these work.
Author Response
Reviewer #2:
The paper could be published, but the reviewer did not understand yet hoy osteocalcin serum level could be modified in these work.
Response: Thank you for your comment. However, there are many studies looking at osteocalcin in serum. Son et al., Paz et al., and Liu et al. use the osteocalcin serum level for study. [1-3] So, it is meaningful to look at serum osteocalcin level in our study.
- Son, S.; Yoon, S.; Kim, M.; Yun, X. Activin A and BMP chimera (AB204) induced bone fusion in osteoporotic spine using an ovariectomized rat model. Spine J. 2020 May;20(5):809-820.
- Paz, L.; Falco, V.; Teng, N.; Reis, L.; Pereira, R.; Jorgetti, V. Effect of 17ß-estradiol or alendronate on the bone densitometry, bone histomorphometry and bone metabolism of ovariectomized rats. Brazilian J. Medical and Biological research (2001) 34: 1015-1022
- Liu, H.; Zhang, H.; Fan, H.; Tang, S.; Weng, J. The preventive effect of Cuscutae Semen polysaccharide on bone loss in the ovariectomized rat model Biomed Pharmacother. 2020 Oct:130:110613.